# Sustainability and Industry 4.0: Definition of a Set of Key Performance Indicators for Manufacturing Companies

**Giuditta Contini** * and **Margherita Peruzzini** 

Department of Engineering—"Enzo Ferrari", University of Modena and Reggio Emilia, Via Vivarelli 10, 41125 Modena, Italy
* Correspondence: giuditta.contini@unimore.it

**Abstract:** Today, sustainability represents a fundamental concept to be developed and implemented in any industrial context. Therefore, it is essential to be able to measure sustainability performance by proper indicators, along the entire lifecycle and the value chain, considering environmental, economic, and social impacts. Moreover, every manufacturing company should have a specific measuring framework to calculate all the specific parameters. In this direction, the modern digital transition and Industry 4.0 (I4.0) technologies are proposing to transform human–machine relations, with a significant impact on social and organizational aspects. At the same time, digitization can help companies to define and implement sustainability by correlating production with proper evaluation metrics. The aim of this research is to provide a complete overview of sustainability Key Performance Indicators (KPIs) based on the Triple Bottom Line concept, referring to the three sustainability areas. Such an overview can be used by companies to set their specific KPIs and metrics to measure their sustainability level, according to their needs.

**Keywords:** sustainability; key performance indicator; digital transformation; Industry 4.0; smart manufacturing; triple bottom line

## 1. Introduction

Sustainability is defined as "development that satisfies the needs of the present without compromising the ability of future generations to satisfy their own needs" [1]. Consequently, designing in a sustainable way means implementing a strategic plan to be increasingly competitive in the market. Presently, sustainability is a crucial concept that is considered very important when you want to design in an ecological, economic, and socially-respectful way. In this direction, the adoption of reliable systems to measure and quantify sustainability is fundamental for companies to understand their position according to modern sustainability goals, to strengthen their competitiveness, and to make sustainability a critical factor for their success [2].

In this context, three dimensions of sustainability have been defined in the literature [3]: environmental, economic, and social. In 1998, Elkington first defined the concept of the Triple Bottom Line (TBL) to treat all three dimensions of sustainability with equal importance [2]. The environmental dimension of the TBL concept is based on the relationship between the use and the renewal of natural resources. Referring to manufacturing industries, this dimension is manifested only in the use of renewable natural resources with zero emissions. This dimension is therefore linked to the concept of recycling and regeneration of resources. The social dimension refers to all those actions that make it possible to better preserve and develop the management of human resources. The third economic dimension refers to the ability to create value with a business strategy capable of balancing costs and revenues. It includes both the management of the economic and financial performance of the industry [4]. In this direction, the assessment of sustainability impact is a mandatory step forward in the achievement and improvement of company sustainability. Being able

to manage the sustainable performance of a company means implementing a sustainable strategy. Despite this, being able to implement this plan still appears to be a significant gap in the literature. Having a set of sustainability performance indicators is a prerequisite for effective performance management [2]. The evaluation and management of corporate sustainability allow for the prevention and avoidance of problems. It is possible to eliminate and reduce risks, conform to standards and regulations, foresee threats, reduce costs, increase efficiency, strengthen competitive advantages, facilitate sustainability reporting and improve operational performance [5]. Every manufacturing company should have a measuring tool to be able to calculate all the specific parameters by proper Key Performance Indicators (KPIs). Lord Kelvin defined KPIs as "When you can measure what you are speaking about and measure it in numbers, you know something about it, when you cannot express it in numbers, your knowledge is of meager and unsatisfactory kind; it may be the beginning of knowledge, but you have scarcely, in your thoughts advanced to the stage of science" [6]. A KPI is a measurable value that demonstrates the effectiveness with which a company is achieving its main business objectives. A KPI is then transformed into a measurable number by proper evaluation metrics. In terms of sustainability, at the company level, a proper set of KPIs and related metrics need to be defined to quantify economic, environmental, and social performance.

Even though in recent years, companies have adopted different types of standards of the International Organization for Standardization (ISO) in their strategic plans, the results show that social and organizational aspects are still considered less important compared to environmental and economic ones. Therefore, implementing a social sustainability plan can be considered fundamental since neglecting fundamental social aspects can cause serious difficulties in showing itself to be a virtuous company [7]. Attention to workers, their working conditions, and gender equality appears to be increasingly important.

In recent years, the social aspects have been a source of studies for numerous researchers who have shown a strong interest in investigating and deepening the social dimension [8]. In 2015, the United Nations Agreement based on 17 Global Goals for Sustainable Development was established. In particular, all aspects not related to the economic sphere, but to the social and environmental one, were included. According to the arguments of the European Commission for sustainable development, the quality of human resources is a key competitive issue, which requires constant attention by management to different dimensions of economic, social, and environmental changes [9].

In addition to the concept of sustainability, the concept of I4.0 has been gaining ground among the main manufacturing companies in recent years. However, many companies are still opposed to turning their manufacturing actions into smart grid-connected processes. In fact, in order to have a smart factory according to the I4.0 concepts, high investments and knowledge are required, which not all companies have. The digital transition of factories and I4.0 technologies have not yet been fully exploited to correlate production and social metrics. As a result, there is a lack of adequate tools for monitoring social and organizational performance in the factory environment. The digitization of production processes not only enables the assessment of environmental and economic impact, but it can also play a key role in knowing the social performance of a manufacturing organization and identifying the social dimension in the circular economy scenario [10].

The combination of I4.0 technologies with the concept of sustainability would make each company a resilient and competitive firm in the market. Despite this, no studies have been found in the literature that define a methodology able to measure and quantify sustainability in a standardized way [7].

The aim of this study is to review the state of the art about sustainability indicators for manufacturing companies and to present a set of indicators that allow the measurement of sustainability according to its three areas, with particular attention to organizational and social impacts.

The rest of the article is divided as follows: chapter 2 illustrates the methodology used to select the articles and to carry out the paper review; chapter 3 investigates the

resulting KPIs and their use in different manufacturing contexts, and finally selects a set of KPIs suitable for manufacturing contexts; chapter 4 discuss the results and finally chapter 5 presents the conclusions.

## 2. Materials and Methods

### 2.1. Research Questions

The aim of this research is to fill the gap found in the research by defining a set of sustainability performance indicators in an industrial context. This goal evokes three research questions, which are examined in this article:

1. How can digitization help to have data in real time and constantly updated?
2. How can one measure a generic set of sustainability KPIs in industrial contexts?
3. How can one refer to the different areas of sustainability in the economic, social, and environmental fields?

### 2.2. Methodology

The research was carried out considering the state-of-the-art sustainability assessment indicators in the last 8 years on an international scientific scenario. We have considered different databases to provide an excellent view of the sustainability indicators used for industrial purposes, across countries and industrial sectors. The literature was identified through the following scientific paper databases: Scopus, Science Direct, Elsevier, IEEE, Google scholar, Taylor and Francis, and Springer.

A structured methodology was adopted to select the papers. We started with the insertion of macro-topics on the databases such as ("KPI" Or "Indicator" Or "Key Performance Indicator" Or "Metric" Or "LCA Indices") And ("Sustainability" Or "TBL" Or "Triple Bottom Line" Or "Sustainable Development") And ("Company" Or "Manufacturing Company" Or "Industry" Or "Firm" Or "Corporate" Or "Manufacturing" Or "Production" Or "Industrial Machines") and, for each filter used, the number of articles was reduced. Terms relating to indicators, methods of selecting these indicators, the concept of sustainability, and the industrial context were used. After this analysis, the number of articles was 12,194.

The analysis was limited by the type of article, considering only the readable papers (selecting the "Open Access", "Gold" and "Access allowed via university database or for a fee"), by year (from 2015 to 2022), by topic ("Environmental Science", "Environmental" and "Energy", "Business, Management and Accounting", "Computer Science", "Decision Sciences", "Economics, Econometrics and Finance", "Engineering"), by type of article ("Journal", "Conference" or "Review") and by language ("English"). Finally, we also searched for further articles by reading the references of the most important selected papers. The selection then led to the analysis of 63 articles.

The bibliographic search generated 12,194 results starting from the selection of the concepts as mentioned above. The large number of articles written testifies to the importance of the topic and the need for a review article that summarizes the latest research and discoveries. The diagram shown in Figure 1 describes in detail the methodology for paper selection.

- *Free access*: we started with 12,194 articles, which were then reduced to 8704 after having selected those with free access ("open access", "gold", "access allowed via university database or for a fee");
- *Year*: we reduced the articles to 5530 after the analysis of the year of publication—only papers published in 2015 and later were included. We selected articles starting from 2015 as it has been shown in previous articles that this year the peak of publications in this area was highlighted [11]. Nonetheless, additional articles (found in the bibliography of other articles) deemed relevant for the analysis were considered;
- *Topic*: 4419 articles were discarded after selecting the main topic of the articles analyzed ("Environmental Science", "Environmental and "Energy", "Business, Management and Accounting", "Computer Science", "Decision Sciences", "Economics, Econometrics and Finance", "Engineering");

- *Type of document*: of the 1111 remaining articles, articles published in journals, conference papers and review papers were selected;
- *Keywords*: to further narrow the search, keywords that are most relevant to the topic were selected, resulting in 175 articles ("Sustainable Development", "Sustainability", "Environmental Impact", "Life Cycle", "Circular Economy", "Indicators", "Sustainability Indicators", "Environmental Indicators", "LCA", "Supply Chains", "Sustainability Assessment", "Ecology", "Environmental Impact Assessment", "Social Sustainability", "Triple Bottom Line", "Sustainability Performance", "Economic Indicators", "Economic Impacts", "Factor Analysis", "Manufacturing Industries", "Sustainable Supply Chain", "Green Economy", "Balanced Scorecard Key Performance Indicators");
- *Language*: English was chosen as the writing language for the article, resulting in 163 articles;
- *Title and abstract*: of the 163 articles, the title and abstract were read, and only the articles dealing with the research topic were selected. The final selection of the articles read was 49.
- *Reading references:* of the 49 selected articles, the respective bibliographies were reviewed and further 14 articles deemed important for the research were selected (although they were published before 2015), due to their high relevance to the research topic.

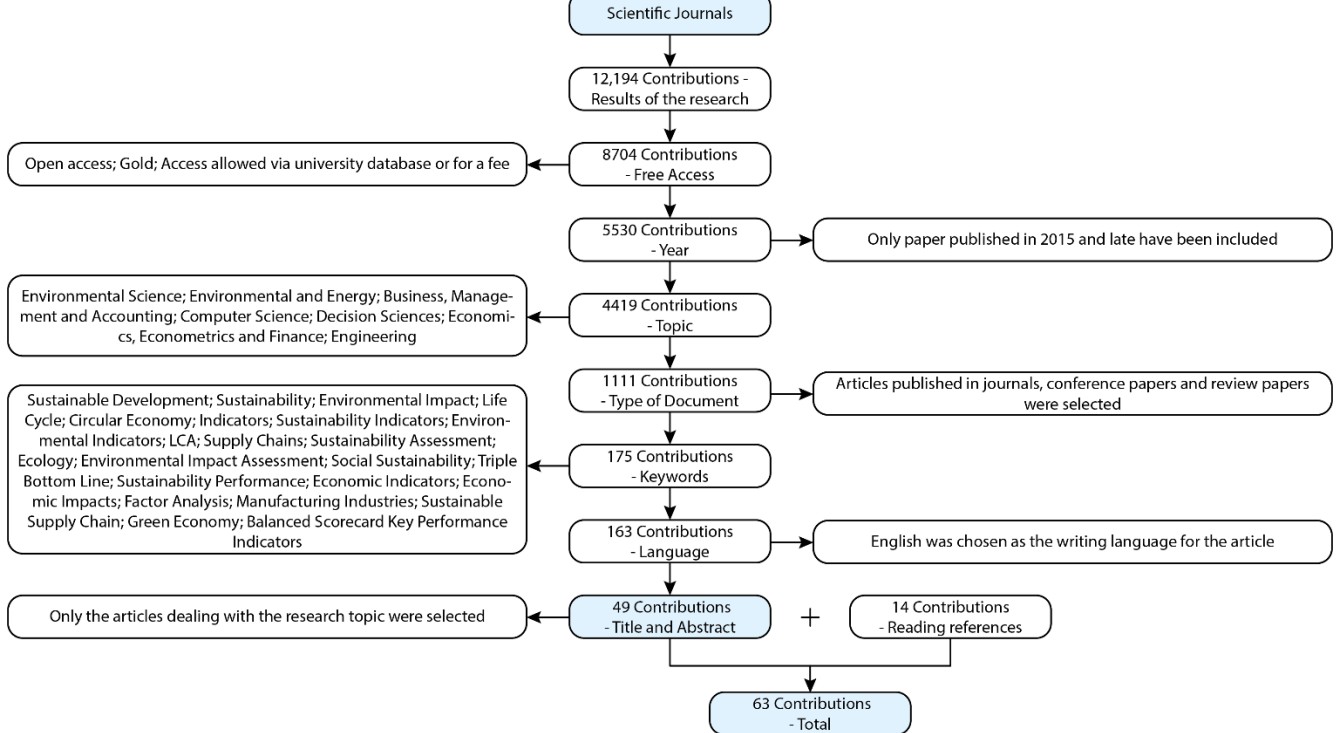

**Figure 1.** Methodology for the review paper selection.

### 2.3. Analysis of the Literature

The analysis of the selected articles allows for a complete overview of what are, to date, the KPIs used in industrial business contexts. To select the set of KPIs, we considered and analyzed the documents containing indicators of all three areas of sustainability. We created a series of KPIs by selecting them from the most recent articles, in order to have an updated state-of-the-art review starting from 2015 to 2022.

Table 1 sums up the main features of the 63 selected articles. For each article, the following was indicated: year of publication, definition of economic, environmental and/or social indicators, metrics for the calculation of KPIs, and the journal in which it was published.

**Table 1.** Selected paper published in different journals and related classes of KPIs (Key Performance Indicators).

| Paper | Year | Eco. | Env. | Soc. | Metrics | Journal |
|---|---|---|---|---|---|---|
| [12] | 2020 | X | X | X | - | Sustainability |
| [13] | 2014 | X | X | X | X | Journal of Cleaner Production |
| [14] | 2014 | X | X | X | - | IEEE International Conference on Industrial Engineering and Engineering Management (IEEM) |
| [15] | 2022 | X | X | X | - | Ecological Economics |
| [16] | 2000 | X | X | X | X | Process Safety and Environmental Protection |
| [17] | 2014 | X | X | X | - | - |
| [18] | 2014 | X | X | X | - | - |
| [19] | 2019 | X | X | X | X | Journal of Cleaner Production |
| [20] | 2017 | X | X | X | - | International Journal of Sustainable Engineering |
| [21] | 2017 | X | X | X | - | Brazilian Journal of Operations & Production Management 14 |
| [22] | 2018 | X | X | X | - | Procedia Manufacturing |
| [23] | 2014 | X | X | X | X | Computers & Industrial Engineering |
| [24] | 2018 | X | X | X | - | - |
| [25] | 2022 | X | X | X | - | Journal of Cleaner Production |
| [26] | 2011 | X | X | X | - | - |
| [27] | 2018 | X | X | X | - | Sustainable Production and Consumption |
| [28] | 2016 | X | X | X | X | Ecological Indicators |
| [29] | 2016 | X | X | X | - | Procedia CIRP |
| [30] | 2011 | X | X | X | X | Ecological Economics |
| [31] | 2019 | - | - | X | - | Remediation Journal |
| [32] | 2019 | X | X | X | - | Resources |
| [33] | 2020 | X | X | X | - | Journal of Cleaner Production |
| [11] | 2018 | X | X | X | - | Journal of Cleaner Production |
| [34] | 2018 | X | X | X | - | Social Sciences |
| [10] | 2021 | - | - | X | - | Journal of Cleaner Production |
| [35] | 2019 | X | X | X | - | Social Sciences |
| [36] | 2020 | X | X | X | - | Bus Strat Env |
| [37] | 2020 | - | X | - | - | Procedia Manufacturing |
| [38] | 2005 | X | X | X | X | - |
| [39] | 2017 | X | X | X | - | - |

**Table 1.** *Cont.*

| Paper | Year | Eco. | Env. | Soc. | Metrics | Journal |
|-------|------|------|------|------|---------|---------|
| [40] | 2018 | X | X | X | X | Journal of Cleaner Production |
| [41] | 2008 | X | X | X | - | Journal of Cleaner Production |
| [8] | 2021 | X | X | X | - | Journal of Cleaner Production |
| [42] | 2018 | X | X | X | - | Nat Resour Forum |
| [43] | 2003 | X | X | X | X | Clean Technologies and Environmental Policy |
| [44] | 2017 | X | X | X | - | Annual Set The Environment Protection |
| [45] | 2015 | - | - | X | - | IEEE Conference on Control Applications (CCA) |
| [46] | 2017 | X | X | X | - | Sustainable Production and Consumption |
| [47] | 2016 | X | X | - | - | MDPI-Sustainability |
| [48] | 2019 | X | X | X | - | Sustainable Development |
| [49] | 2020 | X | X | X | - | Sustainable Production and Consumption |
| [50] | 2020 | X | X | X | - | J. Agile Systems and Management |
| [51] | 2018 | - | - | X | X | - |
| [52] | 2017 | X | X | X | - | Journal of Industrial Information Integration |
| [53] | 2018 | X | X | X | X | Industrial Engineering & Management Systems |
| [54] | 2018 | X | X | X | - | International Journal of Computer Integrated Manufacturing |
| [55] | 2015 | X | X | X | - | International Journal of Productivity and Performance Management |
| [56] | 2018 | - | - | X | X | Journal of Cleaner Production |
| [5] | 2018 | X | X | X | X | Journal of Cleaner Production |
| [57] | 2015 | X | X | X | - | Journal of Cleaner Production |
| [58] | 2019 | X | X | X | X | Sustainable Production and Consumption |
| [59] | 2014 | X | X | X | X | Journal of Industrial Ecology |
| [60] | 2016 | X | X | X | - | Sustainable Development |
| [7] | 2019 | X | X | X | - | Sustainability |
| [61] | 2015 | X | X | X | - | International Journal of Productivity and Performance Management |
| [62] | 2019 | X | X | X | X | Journal of Cleaner Production |
| [63] | 2012 | X | X | X | - | Procedia-Social and Behavioral Sciences |
| [64] | 2014 | X | X | X | - | - |
| [65] | 2020 | X | X | X | - | Journal of Cleaner Production |

**Table 1.** *Cont.*

| Paper | Year | Eco. | Env. | Soc. | Metrics | Journal |
|---|---|---|---|---|---|---|
| [2] | 2019 | X | X | X | X | Journal of Cleaner Production |
| [66] | 2019 | X | X | X | - | JMTM |
| [67] | 2021 | X | X | X | X | Mobile Information Systems |
| [68] | 2022 | X | X | X | X | Journal of Cleaner Production |

Table 1 highlights the large number of articles that list indicators in the three areas of sustainability, but, reading the articles, the authors noted that there is a lack of a single set of generic KPIs to measure sustainability in a manufacturing company. The existing literature does not seem to adequately cover all the pillars of economic, environmental, and social sustainability. Some authors have proposed indicators for measuring industrial performance in different contexts [14,34,48] and only a few have presented all three sustainability indicators [50,52–54]. Furthermore, only a few articles presented metrics capable of quantitatively measuring the reported indicator.

Reading the various articles, we also found the lack of a unique, standardized system to constantly monitor sustainability performance indicators. The purpose of controlling the sustainability parameters is crucial to obtain reliable data and exploit it to one's advantage. Through specific metrics, performance indicators can be identified and measured. For every company, it is fundamental to predict the current sustainability level and to predict the future situation to drive sustainability parameter changes.

## 3. Results

The resulting list of KPIs that emerged from the literature review was reported considering the three areas of sustainability (environmental, economic, and social). Table 2 illustrates all the KPIs broken down by type of industrial sector. All the reported KPIs are with reference to the article in which they were selected. The table shows an exact correspondence of the indicators presented in the respective articles. For the meaning of each indicator, reference is made to the article in which it is mentioned.

### 3.1. Selection of Sustainability KPIs

After the review, a subjective selection of indicators was carried out. Macro-categories were identified for each area. This selection made it possible to group together the main and generic KPIs that can be implemented in any company (regardless of company size and type of production). The selected KPIs are a generic set that must be adapted to every circumstance and type of company. Each KPI can therefore have different metrics with which to be measured depending on the business context.

The KPIs selected are shown below, divided into three categories: social indicators, environmental indicators, and economic indicators.

For each area of sustainability, the KPIs most used by manufacturing industries in recent years have been defined. A clear difference was found in the KPIs available for each sector (economic, environmental, and social). It was found that, only in recent years, the social sustainability indicators have been implemented and applied at an industrial level.

In particular, we focused on social and organizational sustainability indicators due to their importance in modern industrial scenarios but, at the same time, the lack of a standardized evaluation framework. A total amount of 117 indicators were obtained, respectively: 48 social indicators, 30 environmental indicators, and 39 economic indicators.

**Table 2.** KPI of the selected articles.

| Paper | Economic KPIs | Environmental KPIs | Social KPIs |
|---|---|---|---|
| [34] | Manufacturing costs, Commercial costs, Research and development costs, General and administrative costs, Financing costs, Environmental costs, Social costs | Global warming, Ozone depletion, Acidification, Eutrophication, Photochemical ozone, Non-fossil resources, Fossil resources, Raw materials, Consumables, Electrical energy, Thermal energy, Biodiversity, Resource use | Human health, Human resources, Philanthropy |
| [16] | Resource use, Global warming, Ozone depletion, Acidification, Eutrophication, Photochemical ozone, Toxicity, Waste, Material intensity, Energy intensity, Material recyclability, Durability, Service intensity, Voluntary actions, Environmental management systems, Environmental improvements above the compliance levels, Assessment of suppliers | Value added, Contribution to GDP, Expenditure on environmental protection, Environmental liabilities, Ethical investments, Employment contribution, Staff turnover, Expenditure on health and safety, Investment in staff development | Preservation of cultural values, Stakeholder inclusion, Community projects, International standards of conduct, Business dealings, Child labor, Fair trading, Collaboration with corrupt regimes, Intergenerational equity, Income distribution, Employee satisfaction, Satisfaction of social needs, Staff turnover |
| [40] | Cycle time, Changeover time, Uptime, Inventory, Facility costs, Labor costs, Material costs, Utility costs, Net profits, Revenue growth, Return on assets, Profit to revenue ratio, Cost reduction, Adhere to production plan, Improving delivery performance, Energy costs, Direct labor costs, Raw materials costs, Packaging costs, Scrap costs, Consumables costs, Processing tools-related costs, Water costs, Maintenance costs, Cost of PPE, jigs/fixtures, equipment, Other non-operational energy costs, Indirect labor costs, Training costs, Costs of waste disposal treatment, Lead time, Productivity, Utilization of manual labor | Raw materials, Water, Energy, Transportation, Life cycle assessment, Greenhouse gas, Flaring gas, Fresh water used, Oil spills, Waste, Raw materials, Packaging material, Energy, Transportation, Idle energy losses, Renewable energy, Water, Waste, Residue generation intensity, Greenhouse gas, Hazardous gas emission, Material recovered, Consumables recovered, Used packaging material recovered, Used raw material/scrapped parts recovered usage, Hazard materials, Renewable material usage | Physical load index, Noise, Risk, Wage, Workload, Injuries, Injury frequency rate, Social investment, Local procurement and supplier development, Fight against corruption, Workforce diversity and inclusion, Workforce engagement, Workforce training and development, High temperature surfaces, High-speed components and splashes, High-voltage electricity, Physical load index, Work accidents, Work illnesses, Percentage of workers with work-related disease, Noise, Corrosive chemicals, Toxic chemicals, OSHA citations, Employee turnover, Employee satisfaction, Fair trading, Staff training, Diversity, Community quality of life, Community outreach activities, Charitable contributions, Injuries |

**Table 2.** *Cont.*

| Paper | Economic KPIs | Environmental KPIs | Social KPIs |
|---|---|---|---|
| [53] | Manufacturing costs, Commercial costs, Research and development costs, General and administrative costs, Financing costs, Environmental costs, Social costs | Waste, Air emission, Energy, Greenhouse gas, Hazard materials, Ozone, Water, Materials, Energy, Land useBiodiversity, Natural management and conservation | Health and safety, Professional development, Employee satisfaction, Health and safety of the product at use phase, Employee satisfaction, Product responsibility, Fair trading, Equity, Human rights, Public service policy, Justice |
| [32] | Income from sales, Value added, Gross operating profit, Cash flow, Expenses for goods and services, Change in raw mat., consume, goods stock, Use of third parties assets, Cost of human resources, Other operating expenses, Amortization and depreciation, Provisions for risks and others, Financial income and charges, Extraordinary earnings and charges, Taxes, The result of third parties, Total production costs | Energy, Global warming, Toxicity, Air pollution, Ozone, Resources use | Community engagement, Access to resources, Social responsibility, Fair trading, Hours of work, Child labor, Forced labor, Excessive, Working Time, Injuries, Fatalities, Toxics and Hazards, Wage, Justice, Migrant Labor, Collective, Bargaining, etc., Indigenous, Rights, Gender equity, High Conflict, Legal System, Fight against corruption, Drinking Water, Improved Sanitation, Hospital beds, Inadequate social benefits, Labor Rights and Decent Work, Health and Safety, Human rights, Community Infrastructure |
| [66] | Decrease in cost of materials purchased, Decrease in cost of energy consumption, Decrease in fee for waste discharge, Improvement in earnings per share, Improvement in return on investment, Sales growth, Profits growth | Improvement of an enterprise's environmental situation, Waste, Air emission, Hazard materials, Decrease of frequency of environmental accidents, Toxicity | Employee satisfaction, Improvement in its image in the eyes of its customers, Community projects, Improvement in relations with community stakeholders, Improved awareness and protection of the claims and rights of people in community served, Improvement in employee training and education, Improvement in occupational health and safety of employees, Welfare indicators |

**Table 2.** *Cont.*

| Paper | Economic KPIs | Environmental KPIs | Social KPIs |
|---|---|---|---|
| [59] | Capital cost: Equipment cost, Facility cost, R&D cost, Initial training cost, Labor cost, Material cost, Packaging cost, Energy cost, Transportation cost, Warehouse cost, Recovery cost, Product ownership cost, Average disassembly cost, Environmental regulations violation, Other costs related to legal issues, Sales price, Profit, Defective/ returned products loss, Warranty cost | Raw materials, Recycling, Hazard materials, Packaging materials, Recycled packaging material ratio, Material utilization, Regulations and certification: Regulation compliance, Certification, Renewable energy, Coal, Petroleum, Nuclear, Natural gas, Energy regulation compliance, Energy certification, Energy, WaterRatio of recycled water used, Other natural resources used, Natural resource regulation compliance, Natural resource certification, Greenhouse gases, Hazard gaseous emissions, Mass of solid waste landfilled, Reused/recycled hazardous waste, Disposed hazardous solid waste, To hydrosphere, Reused/recycled liquid waste, Disposed hazardous liquid waste, Heat, Noise, Light, Radioactive emissions, Waste management regulations/certification: Waste management regulation compliance, Waste management certification, EOL product, Ease of product disposability, Product disassemblability, Ratio of EOL product recovered, Product reusability, Ratio of EOL product reused, Product remanufacturability, Product redesign, Ratio of product remanufacturedEOL product recycling: Product recyclability, Ratio of product/material recycled, EOL regulation compliance, EOL certification | Reparability, Maintainability, Failure rate, Lifespan, Return rate for product defects, Product recall rate, Major product specifications, Product customizability, Product functional effectiveness, Ease of operation, Ease of EOL product disposal for the user, Ease of EOL product recovery, Product EOL societal impact, Injury rate, Product safety specifications, Health, Product EOL regulation compliance, Product EOL certification, Noise |

**Table 2.** *Cont.*

| Paper | Economic KPIs | Environmental KPIs | Social KPIs |
|---|---|---|---|
| [19] | Investments: R&D investment, Pollution prevention and control investment, Environment investment, Energy efficiency investment, Safety investment, Community investment, Ethics/philanthropy investment, Operating cost, Overhead cost, Packaging cost, Production cost, Set up cost, Inventory cost, Labor cost, Unit cost, Maintenance cost, Taxes, EHS fines, Sales, Market share, Revenues, Profit, Turnover, Throughput, New products, Lead time, Scrap, Quality, Mix flexibility, Volume, Flexibility, DFx, Green product, IT levelSuppliers: Number of suppliers, Local suppliers, Certified suppliers | Water, Total water use, Fresh water use, Recycled water use, Quality of water, Raw materials, Total material use, Recycled material use, Hazardous material use, Toxic material use, Energy, Total energy use for production, Renewable energy use for production, Fuel use for production, Gas use for production, Coal use for production, Total energy use not for production, Renewable energy use not for production, Fuel use, Gas use not for production, Coal use not for production, Air emissions, $CO_2$, Other GHG, NOx, $SO_2$, OD, S Metal emissions, Other emissions, Toxic emissions, Waste, Hazardous solid waste, Non-hazardous solid waste, Hazardous liquid waste, Non-hazardous liquid waste, COD, BOD, Waste water, Chemical waste, Waste disposed, Waste recycled, Energy recovery, Material recovery, Environmental management, Environmental accidents, Environmental fines, Environmental certification, Cost of compliance | Community complaints, Community projects, Local employment, Involvement of local community, Employee satisfaction, Personalized products, Services offered, Number of employees, Wage, Employee satisfaction, Involvement of employees, Gender equity, Discrimination, Safety training, Environmental training, Staff turnover, Work accidents, Injuries, Fatalities, Near misses, PPE, Absenteeism, Noise, Dust, Toxic substances, OHS Administration Citations, Safety expenditure |
| [65] | Electricity cost, Operating and maintenance costs, Cost of capital, Raw Material Cost, Production Cost, Net present value (NPV), Labor cost, Life cycle Cost (LCC), External environmental cost, Profitability, Gross domestic product (GDP) | Eutrophication, Acidification, Global warming, Photochemical ozone, Ozone, Soil occupation and land use, Air emissions, Water, Toxicity, Climate change, Terrestrial ecotoxicity, Human health, Protection and quality of the ecosystem, Abiotic depletion | Employee, Community Health, Occupational, Health and safety, Contribution to economic and technological, Discrimination, Wage, Child labor, Staff training, Working hours, Forced labor, Supplier relations, Community Involvement, Company commitments to environment, Welfare indicators, Philanthropy |
| [5] | Quality, Efficiency, Cost, Timeliness | Air emissions, Water, Land, Energy | Noise, Health, Empl. Satisfa, Custo. Satisfa |

**Table 2.** *Cont.*

| Paper | Economic KPIs | Environmental KPIs | Social KPIs |
|---|---|---|---|
| [28] | Total costs, Staff costs, Operating expenses, Investments, Return on investment, Profit, Sales, Return on sales, Economic value added, Added value, Turnover, Cash Flow, Market share, Return on equity, Return on assets, Return on invested capital, Liquidity, Turnover of assets, Turnover of inventory, Turnover of receivables, Turnover of liabilities, Debt ratio, Reliability of suppliers, Monetary value of sanctions, Expenditure on research and development | Energy consumption, Consumption of energy from renewable source, Consumption of materials and raw materials, Consumption of recycled materials and raw materials, Fuel use, Water, The amount of occupied land, Waste production, Production of hazardous waste, Amount of recyclable waste, Amount of discharged waste water, Greenhouse gas, Emissions of $SO_2$, NOx and particulate matter (PM), Compliance with legal requirements: Number of violations of statutory rules and regulations relating to the environment, Monetary value of fines for violations of laws related to the environment, Environmental investments, Environmental costs | Number of complaints received from the community, Equal opportunities, Discrimination, Wage, Human rights, Staff turnover, Staff trainingEmployee turnover, Percentage of employees covered by collective agreement, Labor relationshipEthics indicators, The overall accident rate, Accident rate, Fatalities, Occupational diseases, The rate of absence, Work accidents, Health and safety of customers, Percentage of products and services for which the impact on the health and safety of customers is evaluated during their life cycle, Employee satisfaction |
| [22] | | Emission, Effluent, Waste, Air emission, Waste Energy Emission, Pollution, Hazard materials, Greenhouse gas, Ozone, Other Pollutants, Water, Raw materials, Energy, Land use, Biodiversity, Habitat Management, Conservation | Health and Safety, Development, Employee satisfaction, Health and Safety, Employee satisfaction, Customer Rights, Product responsibility, Justice, Development |
| [13] | Environmental costs, Buying environmentally friendly materials, Employee satisfaction, Customer returns, Risks and recoverability, Net life cycle cost, Long-term debt, including current portion, Returning customers ratio, Level of supplier preprocessing of raw materials, Cash flow, Cash flow provided by operating activities, Cooperation degree, Networks, Profit, Market share, Sales, Existing efficiency vs. cost of upgrading, Increased cost efficiency, Cost savings, Operational performance | Environmental social concerns, Cooperation with customers for green packaging, Risk of severe accidents, Environmental risks, Life cycle assessment (LCA), Energy, Environmental partnership with suppliers, Choosing suppliers according to environmental criteria, Annual mass-flow of different materials used, Collaborating with other companies and organizations for environmental initiatives, Improving opportunities for reducing waste through cooperation with other actors, Interaction and harmony co-exist with natural systems on production and consumption systems, Energy requirement per unit of net value added, Global warming, Energy efficiency, Recycling, Waste, Air emission | Voluntary programs, Number of individual volunteering, Risk, Health status and risks, Community engagement, Stakeholder empowerment, Labor relationship, After sales service, Publicly available missions and values statement, Value added and community benefits, Institutional efficiency, Noise, Health and safety performance measurement systems |

**Table 2.** *Cont.*

| Paper | Economic KPIs | Environmental KPIs | Social KPIs |
|---|---|---|---|
| [64] | Supply chain cost, Service level | Greenhouse gas, Water, Energy, Waste, Hazard materials, Toxicity | Labor practices and decent work, Human rights, Society, Product responsibility |
| [62] | Supply chain management, IT, Energy price, Emerging markets, Business models, Process technology, Government regulations, Growth of population, Growth of economics, Consumption of resources, Needs, Market opportunity, Product development cost, Product development time, Development capability, Regionalized products, Personalized products, Enterprise size, Enterprise functionality, Material handling equipment, Material handling storage, Identification system, Plant location, Functional layout, Product layout, Cellular layout, Complexity analysis, Lean production, Agile manufacturing, Remanufacturing, Recycling processes, Product costs, Response, Enterprise productivity, HR appraisal, Resources status, Product quality, Strategic planning, Organizing work, Organization structure, Leadership role, Staffing, Managing culture | Environmental budget, Environmental certification, Environmental concerns and compliance, Workers implications, Use of resources, Renewable energy, Recycling, Recycled water, Recyclable wastes, Pollution, Air pollution, Water pollution, Land pollution, Dangerous inputs, Dangerous outputs, Dangerous wastes, Eco-system services, Biodiversity, Land use, Development of rural areas | Employment, Work conditions, Social dialogue, Society security, HR development, Human rights, Child labor, Freedom of association, Discrimination, Societal commitment, Involvement in local community, Education, Healthcare, Job creation, Societal investment, Culture and technological development, Marketing and information, Protection of private life, Access to resources, Fight against corruption, Fair trading, Understanding foreign culture |
| [18] | Turnover, Operational profit, Investments, Net profit, Research and development expenses | Natural gas consumption, Energy, Water, Carbon dioxide emissions, Nitrogen oxide emissions, Sulphur oxide emissions, Water, Water suspensions, Hydrogen sulfide, Biological oxygen demand, Chemical oxygen demand, Water temperature, Chromium, lead, copper, nickel in water, Phosphorus, Abluents, Salvaged waste, Waste | Work accidents, Work illnesses, Staff training, Non-profit programs, Staff turnover |
| [35] | Environmental LCC, Conventional LCC, Electrical energy, Thermal energy, Consumables | Raw materials, Fired waste milled, Respiratory inorganics, Land use, Aquatic eutrophication, Global warming, Dimensional quality, Water, Ecosystem services, Access to water, Biodiversity | Societal LCC, Human resources, Human health |

**Table 2.** *Cont.*

| Paper | Economic KPIs | Environmental KPIs | Social KPIs |
|---|---|---|---|
| [12] | % of income for recycling programs, Investment in technology rate, % of production sites with an environmental certificate | Greenhouse gas, Carbon footprint rate, % of waste generated per thousand products units, Recycling, Energy, Water, Renewable energy, Sustainable water use rate | Employee satisfaction, Staff training, Health and safety rate, Diversity, Equality rate, Total expenses for social initiatives, Work illnesses, Labor relationship |
| [21] | Employment, Work conditions, Respect of social dialog, Health and security, Development of human resources, Human rights, Child labor, Forced labor, Freedom of association, Discrimination, Involvement in local communities, Education, Culture and technological development, Job creation, Healthcare, Social investment, Customer issues, Marketing and information, Healthcare and security, Protection of private life, Access to essential services, Business practices, Fight against corruption, Fair trading, Promotion of corporate social responsibility in the sphere of influence | Environmental budget, Environmental certification, Environmental compliance, Worker implications, Use of resources, Renewable energy, Recycling, Inputs from recycling, Recyclable outputs, Recyclable wastesPollution: Air pollution, Water pollution, Land pollution, Other pollution, Dangerous inputs, Dangerous outputs, Dangerous wastes, Natural environment, Eco-systemic services, Biodiversity, Land use, Development of urban and rural areas | Customer service, Supplier service, Reliability of stocks, Reliability of estimates, Design responsiveness, Purchase responsiveness, Source responsiveness, Production responsiveness, Delivery responsiveness, Sales responsiveness, Return responsiveness, Supply chain responsiveness, Product responsibility, Flexibility of suppliers, Supply flexibility, Production flexibility, Delivery flexibility, Financial performance, Design cost, Purchase cost, Source cost, Production cost, Delivery cost, Return cost, Supply chain cost, Product/ service quality, Quality performance of suppliers, Production quality |
| [2] | Economic performance, Investments in new processes and R&D, Economic value distributed, Direct economic value generated, Environmental investments | Raw materials, Energy, Water, Greenhouse gas | Employee turnover, Occupational health and safety, Lost time injury frequency rate, Staff training |
| [30] | Innovations created through supplier partnerships, The number of stores, Total space, Total sales/kTRY, The number of people employed, Total tax paid/kTRY, The number of shareholders, Establishing new employment | Annual water consumption, Annual energy consumption, Waste, Fraction of suppliers certified in ISO 14001, Number of ISO standards developed, Fraction of facilities using HFC-powered units, Fraction of facilities using renewable energy, Recycling, Effectiveness of reverse logistics system, | Staff training, Number of applied innovative ideas generated by employees per employee per year, Staff turnover, Annual number of recordable incidents with respect to harassment and violence per employee, Work accidents, Average annual areic number of recordable employee complaints per employee, |

**Table 2.** *Cont.*

| Paper | Economic KPIs | Environmental KPIs | Social KPIs |
|---|---|---|---|
| | opportunities, Promoting new investments, Competitiveness of reverse and forward logistics system | Effectiveness of the 3PL company that the company works with, Effectiveness of supplier training in environmental issues, Effectiveness of supplier monitoring | Average annual areic number of customer complaints, Gender equity, Equal employment opportunities, Community projects, Effectiveness of discipline management, Effectiveness of compensation management, Effectiveness of personnel recruitment and selection, Organization's openness to stakeholder involvement in decision making, Institutional efficiency, Effectiveness of performance management system |
| [23] | Customer service, suppliers' service, reliability of stocks, reliability of forecastsResponsiveness: Design responsiveness, purchase responsiveness, source responsiveness, production responsiveness, delivery responsiveness, sell responsiveness, return responsiveness, supply chain responsiveness, Suppliers flexibility, supply flexibility, production flexibility, delivery flexibility, Financial performance, Design cost, purchase cost, source cost, production cost, delivery cost, return cost, supply chain cost, Product/service quality, quality performance of suppliers, production quality | Environmental management, Environmental budget, environmental certification, Environmental compliance, workers implications, Renewable energy, recycled water, Recycling, Inputs stemming from the recycling, Recyclable outputs, Recyclable wastes, Pollution, Air pollution, Water pollution, Land pollution, other pollution, Dangerous inputs, dangerous outputs, dangerous wastes, Eco-systemic services, respect of biodiversity, land use, development of urban and rural areas | Employment, Work conditions, Respect of social dialog, Health, Security, Human resources, Human rights, Child labor, Forced labor, Freedom of association, Discrimination, Involvement in local community, Education, Culture and technological development, Job creation, Societal investment, Marketing and information, Health, Security, Protection of private life, Access to resources, Fight against corruption, Fair trading, Social responsibility |
| [55] | On-time delivery, Employee satisfaction, Order fill rate, Product/service availability, Distribution costs, Total costs, Transport costs, Loading capacity utilization, Stock-outs, Product lateness, Lead time, Forecast accuracy | Air emission, Level of $CO_2$ emission, Level of $CO_2$ emission from transport processes, Level of $CO_2$ emission from infrastructure, Natural resources utilization, Energy, Water, Energy consumption/revenue, Waste, Recycling, Level of waste, Level of products recycled, Level of products reused | Health and safety, Work accidents, Work conditions, Number of accidents, Noise, Noise volume, Time of noise emission, Noise emission in urban areas, Employees skills, Employee satisfaction, Percent of labor cost spent on training |

**Table 2.** *Cont.*

| Paper | Economic KPIs | Environmental KPIs | Social KPIs |
|-------|---------------|--------------------|--------------|
| [61] | Cost, Capacity utilization, Cost variance from expected costs Inventory levels, Labor efficiency, Supplier cost-saving initiatives, Time, Amount of goods delivered on-time, Efficiency of purchase order cycle time, Efficiency of the production lines Information timeliness, Percentage of late deliveries, Purchase order cycle time, Supplier lead time against industry norm, Supplier's booking-in procedures, Buyer-supplier partnership level, Delivery reliability, Distribution of decision competences between supplier and customer, Extent of mutual assistance leading in problem-solving efforts, Extent of mutual planning cooperation leading to improved quality, Information accuracy, Information availability, Level of supplier's defect-free deliveries, Mutual trust, Percentage of wrong supplier delivery, Quality and frequency of exchange of logistics information between, Quality of perspective taking in supply networks, Satisfaction with knowledge transfer, Satisfaction with supplier relationship, Supplier and customer, Supplier assistance in solving technical problems, Supplier rejection rate, | Dependence on imports, Dependence on imports of solid fuel, Dependence on natural gas imports, Dependence on oil imports, Differentiation of energy fuel, Differentiation of fuel of electrical energy production, Energy, Differentiation of primary fuel, Process modifications, Publicly available missions and values statements, Raw materials, Source reduction activities, Strategic oil supplies, Fuel use, Adjustment of energy pricelist, Dividing of public enterprise, Efficiency of electrical energy production, Efficiency of energy conversion, Energy intensity, Energy law for the reforming and privatization of energy enterprises, Habitat improvements and damages due to enterprise operations, Independent energy regulator, Level of competition, Major awards received, Per capita electrical energy consumption, Per capita energy consumption, Per capita fuel consumption, Per capita fuel consumption, Private participation, Quantity of non-product output returned to process or market by recycling or reuse, Total electrical energy consumption, Total energy consumption, Total fuel consumption, Total water consumption, Transformation of energy sector, Application of Kyoto protocol, Emitted $CO_2$ per capita, Emitted $CO_2$ per electricity and steam production, Emitted $CO_2$ per GDP, Emitted $CO_2$ per gross domestic energy consumption, Environmental liabilities under applicable laws and regulations, Formal, written commitments requiring an evaluation of life cycle impacts, Indicators of intensity of emitted $CO_2$, Non-production releases, On-site and off-site energy recovery, On-site and off-site recycling, | Existence of equal opportunity policies or programs, Percentage of senior executives who are women, Percentage of staff who are members of visible minorities, Percentage of staff with disabilities, Diversity, Percentage of employees represented by independent trade union organizations or other bona fide employee representatives, Percentage of employees covered by collective bargaining agreements, Number of grievances from unionized employeesHealth and Safety, Evidence of substantial compliance with international labor organization guidelines for occupational health management systems, Number of workplace deaths per year, Existence of well-being |

**Table 2.** *Cont.*

| Paper | Economic KPIs | Environmental KPIs | Social KPIs |
|---|---|---|---|
| | Materials variety, Product and service variety, Product development time, Product volume variability capabilities, Response to product changes, Supplier ability to respond to quality problems, Involvement in new product design, Introduction of new processes, Satisfaction with knowledge transfer satisfaction, Technological capability levels | On-site or off-site treatment, Percentage of renewable energy sources in the electrical energy production, Percentage of renewable energy sources in the primary energy production, Procedures to assist product and service designers to create products or services with reduced adverse life cycle impact, Programs or procedures to prevent or minimize potentially adverse impacts of products and services, Air emission | programs to encourage employees to adopt healthy lifestyles, Percentage of employees surveyed who agree that their workplace is safe and comfortable, Child labor, Number of children working, Whether contractors are screened for use of child labor, Child labor, Percentage of pre-tax earnings donated to the community, Involvement and/or contributions to projects with value to the greater community, Existence of a policy encouraging use of local contractors and suppliers |
| [63] | Innovations created through supplier partnerships, Total sales, The number of shareholders, Promoting new investmentsCustomer complaints, Equal employment opportunities, Total tax paid, Competitiveness of the forward and reverse supply chain sub-criteria are used to evaluate the sustainable economic performance | Waste, Number of ISO standards developed, Renewable energy, Effectiveness of reverse logistics system, Effectiveness of supplier training in environmental issues, Fraction of suppliers certified in ISO 14001, Fraction of facilities using HFC powered units, Recycled materials, Effectiveness of the 3PL company, The number of stores, Energy, Water, Sub-criteria used to evaluate the sustainable resource performance | Staff training, Applied innovative ideas generated by employees, Staff turnover, Recordable incidents with respect to harassment and violence/employee, Work accidents, Recordable employee, Customer complaints, Community projects, Effectiveness of discipline management, Effectiveness of compensation management, Effectiveness of Personnel, Recruitment and Selection, Organization's openness to stakeholder involvement in decision-making, Institutional efficiency, Effectiveness of performance management system subcriteria used to evaluate the sustainable social performance |
| [43] | Packaging costs, Investments in sustainable development, Investments in environmental protection, Investments in ethical activity, Complaints of customers, Fraction of suppliers, Cost of employee, Costs of health protection of employee | Energy, Renewable energy, Energy for recycling, Raw materials, Renewable raw materials, Hazard materialsWater, Durability, Waste, Liquid and Solid waste, Pollution, Greenhouse gas, Acidification, Photochemical ozone, Eutrophication, Noise, Air emission | Employee number, Employee turnover, Payment ratio, Employee satisfaction, Work illnesses, Community projects, Community population growth, Noise |

**Table 2.** *Cont.*

| Paper | Economic KPIs | Environmental KPIs | Social KPIs |
|---|---|---|---|
| [14] | Inventory cost, Labor cost, Material cost, Product delivery, Raw material substitution | Air emission, Energy, Fuel use, Land use, Material consumption, Raw materials, Noise, Nonproduct output, Water | Work accidents, Employee involvement, Gender equity, Occupational health and safety, Staff training, Noise |
| [22] | | Emission, Solid waste, Waste, Air emission, Waste energy emission, Hazard materials, Greenhouse gas, Ozone depletion, Pollutants, Water, Material, Raw materials, Energy, Land use, Biodiversity, Habitat management, Conservation | Health and safety, Development, Employee satisfaction, Employee satisfaction, Customer rights, Product responsibility, Justice |

### 3.1.1. Social KPIs

A focus was placed on social and organizational sustainability indicators as, only recently, social, and organizational sustainability concepts have been implemented in corporate contexts. The monitoring of these aspects appears to be fundamental for a successful business strategy.

Figure 2 illustrates the selected 48 social KPIs and how many times each indicator appeared in the articles analyzed in the review. Results show higher attention to "health", used by 32 papers. Below are: "employee satisfaction", "staff training", "noise", and "work accidents". These parameters are those that companies currently consider the highest priority. Table A1 in Appendix A lists the articles presenting each macro-category of selected social indicators.

### 3.1.2. Environmental KPIs

The environmental indicators are those which, together with the economic indicators, are the most used in industrial contexts. Figure 3 illustrates the 30 selected environmental KPIs and how many times each indicator appeared in the articles analyzed. As shown in Figure 3, the most widely used environmental indicator is "energy", cited by 38 papers, followed by: "waste", "water", and "raw materials". Table A2 in Appendix A lists the articles presenting each macro-category of selected environmental indicators.

### 3.1.3. Economic KPIs

Economic indicators are indicators that have always been used in every business context. In fact, they are fundamental as they allow us to understand the company's performance and to be able to implement initiatives to get more company revenue. Figure 4 illustrates the 39 selected economic KPIs and how many times each indicator appeared in the articles analyzed. Results show that the most important indicator is "turnover", cited by 15 papers, followed by "material costs", "quality", "sales", and "labor costs". Table A3 in Appendix A lists the articles presenting each macro-category of selected economic indicators.

### 3.1.4. Set of KPIs Selected

After selecting and grouping the indicators into macro-categories, a comprehensive overview was defined. Table 3 shows the set of KPIs created, capable of monitoring sustainability in any industrial context. This set can be used by any manufacturing company that wants to design in a sustainable way or assess their processes according to the sustainability viewpoint.

## Social Indicators

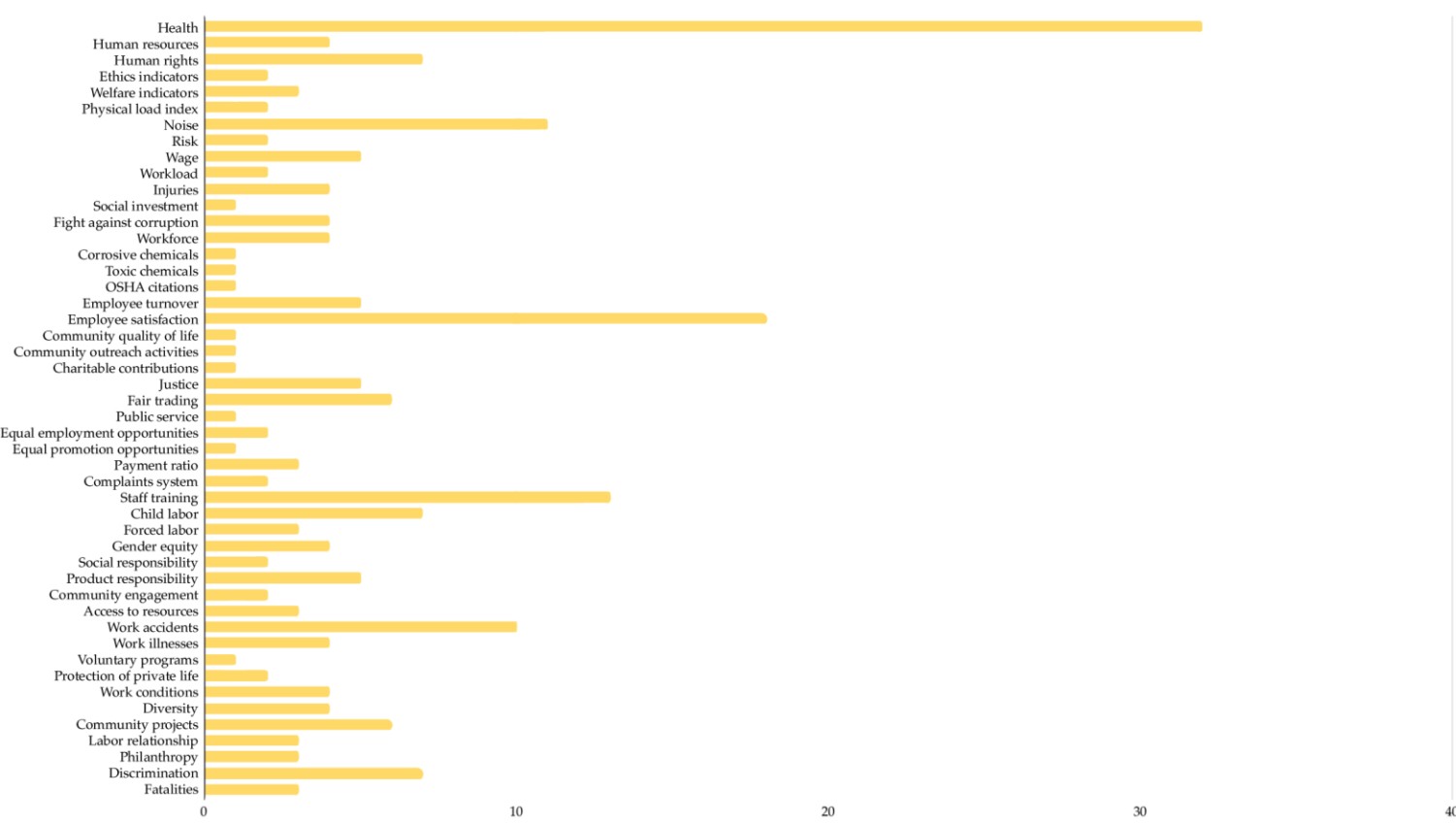

**Figure 2.** Social Indicators.

## Environmental Indicators

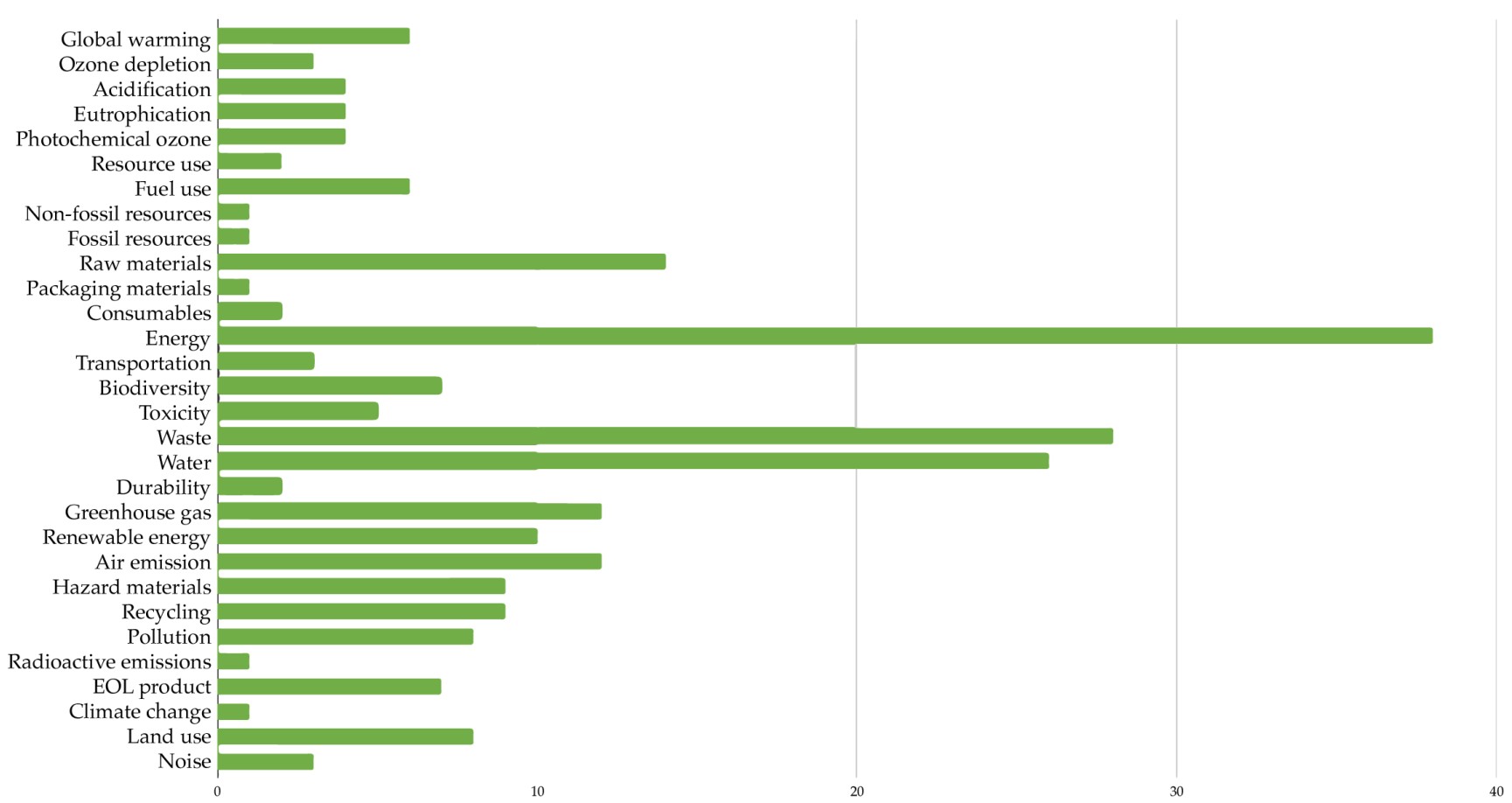

**Figure 3.** Environmental Indicators.

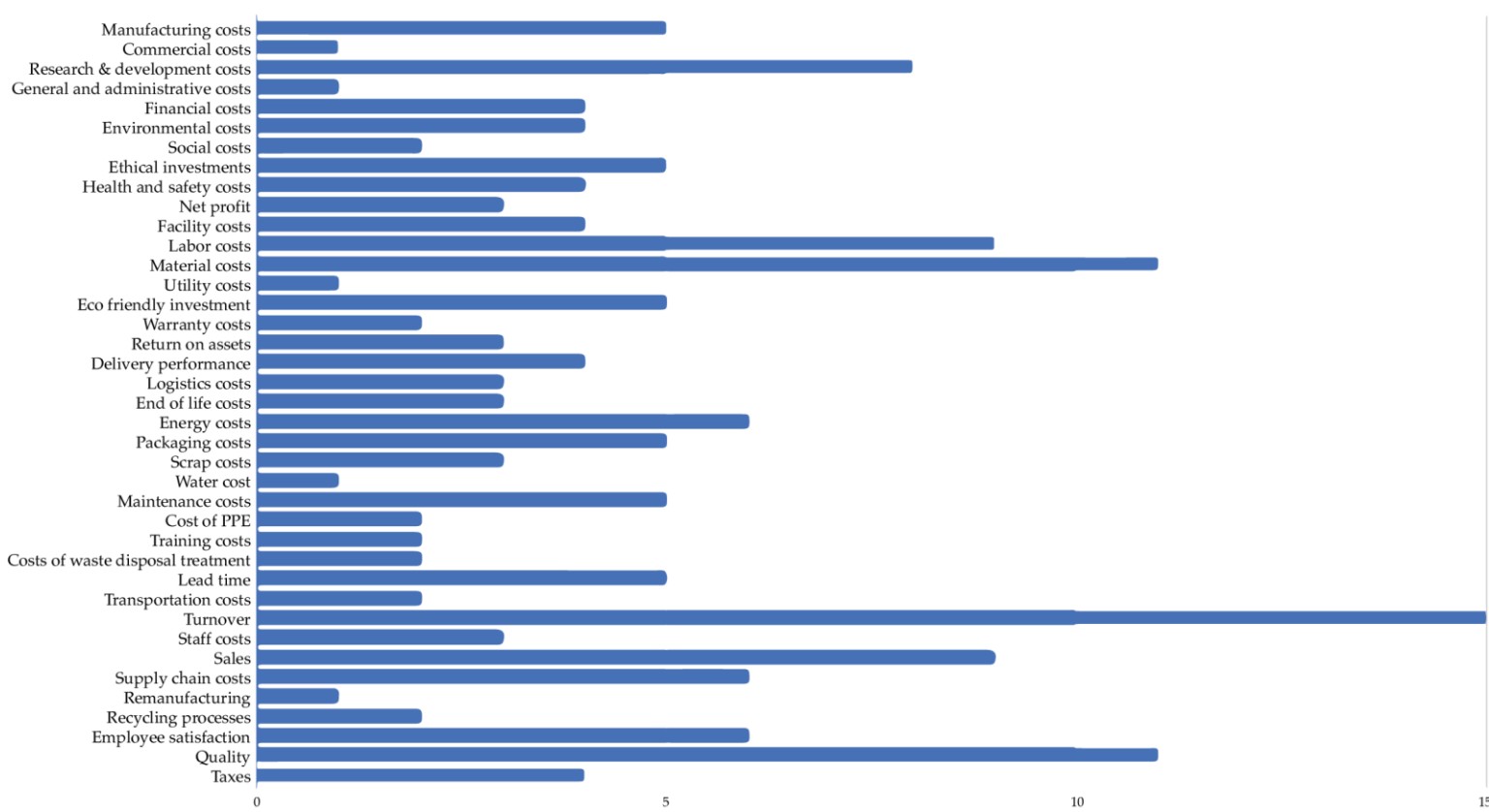

**Figure 4.** Economic Indicators.

**Table 3.** Set of KPIs selected.

| Social KPIs | Environmental KPIs | Economic KPIs |
|---|---|---|
| Health | Global warming | Manufacturing costs |
| Human resources | Ozone depletion | Commercial costs |
| Human rights | Acidification | Research and development costs |
| Ethics indicators | Eutrophication | General and administrative costs |
| Welfare indicators | Photochemical ozone | Financial costs |
| Physical load index | Resource use | Environmental costs |
| Noise | Fuel use | Social costs |
| Risk | Non-fossil resources | Ethical investments |
| Wage | Fossil resources | Health and safety costs |
| Workload | Raw materials | Net profit |
| Injuries | Packaging materials | Facility costs |
| Social investment | Consumables | Labor costs |
| Fight against corruption | Energy | Material costs |
| Workforce | Transportation | Utility costs |
| Corrosive chemicals | Biodiversity | Eco friendly investment |
| Toxic chemicals | Toxicity | Warranty costs |
| OSHA citations | Waste | Return on assets |
| Employee turnover | Water | Delivery performance |
| Employee satisfaction | Durability | Logistics costs |
| Community quality of life | Greenhouse gas | End of life costs |
| Community outreach activities | Renewable energy | Energy costs |
| Charitable contributions | Air emission | Packaging costs |
| Justice | Hazard materials | Scrap costs |
| Fair trading | Recycling | Water cost |
| Public service | Pollution | Maintenance costs |
| Equal employment opportunities | Radioactive emissions | Cost of PPE |
| Equal promotion opportunities | EOL product | Training costs |
| Payment ratio | Climate change | Costs of waste disposal treatment |
| Complaints system | Land use | Lead time |
| Staff training | Noise | Transportation costs |
| Child labor | | Turnover |
| Forced labor | | Staff costs |
| Gender equity | | Sales |
| Social responsibility | | Supply chain costs |
| Product responsibility | | Remanufacturing |
| Community engagement | | Recycling processes |
| Access to resources | | Employee satisfaction |
| Work accidents | | Quality |
| Work illnesses | | Taxes |
| Voluntary programs | | |
| Protection of private life | | |
| Work conditions | | |
| Diversity | | |
| Community projects | | |
| Labor relationship | | |
| Philanthropy | | |
| Discrimination | | |
| Fatalities | | |

## 4. Discussion

The aim of the research was to classify all the KPIs currently cited by the recent scientific literature and used in industrial contexts, divided into the three areas of sustainability (environmental, economic, and social). As a result, it can be stated that the current literature can provide a "flat" set of KPIs without any specific guideline to implement them in companies, considering the different sectors, different sizes, and types of production.

As shown by the high number of KPIs present in the literature, every company wants to be able to measure its sustainability performance. Applying these performance

indices, the company acquires value in terms of visibility and virtuosity of industrial activities. Being sustainable currently means being able to have a competitive business model integrated with a corporate strategic plan that incorporates the monitoring and management of sustainability parameters.

At the beginning of the article, the authors asked themselves a series of questions that this research tried to answer. These research questions are:

1.  How can digitization help to have data in real time and constantly updated?
2.  How can one measure a generic set of sustainability KPIs in industrial contexts?
3.  How can one refer to the different areas of sustainability in the economic, social, and environmental fields?

To answer the first question "How can digitization help to have data in real time and constantly updated?" we must take into consideration the new technologies of I4.0. Thanks to the complete digitalization of production, the integration of digital technologies into business processes, and good management of big data, environmental, economic, and social monitoring can be carried out dynamically and in real time. Eco-design, in a simulation environment, makes it possible to predict the environmental, social, and economic performance of alternative industrial solutions. IoT technologies make it possible to measure the effects in real time as they occur, providing the ability to intervene in processes to mitigate them. Today, I4.0 technologies are actively used to optimize production operations with related capabilities. Once active, digital tools allow manufacturers to optimize efficiency, translate data into actionable information, and produce more accurate demand forecasts and predictive maintenance programs. These data are provided by performance indicators that make it possible to receive the necessary data on which to make predictions and assessments.

The second question research was "How can one measure a generic set of sustainability KPIs in industrial contexts?".

Thus, the review provides a set of indicators able to measure the overall company sustainability performance in different industrial contexts and with different production processes. We have developed a generic set of KPIs to assess sustainability performance in the industrial chain. This set of indicators will have to be flexible and adaptable to any industrial context in order to have a starting point for key indicators. Each company will then have the task of implementing specific KPIs according to their situation. In fact, for the generic set of KPIs selected, you can have numerous evaluation metrics depending on the business context. In this way, each company can be oriented towards the calculation of sustainability starting from a standardized methodology.

To answer the third question "How can one refer to the different areas of sustainability in the economic, social, and environmental fields?", we have analyzed the state of the art in the literature, and we have presented all the indicators that are currently used in industrial contexts. These indicators have been divided into the three categories of the Triple Bottom Line. For each area (social, environmental, and economic), macro-categories of KPI have been identified in order to individuate general indicators that can be applied in any single company.

For each KPI, a set of metrics need to be defined according to the specific context of application. For instance, "health" can be measured by some metrics such as: percentage of workers with work-related disease, percentage of employees receiving safety training, percentage of workstations with high voltage electricity, percentage of workstations with corrosive/toxic chemicals, physical load index, work-related injuries and illness, number of accidents per year. Diversely, "employee satisfaction" can be measured by: ratio between the number of sickness absence rate in an entity and the national average sickness absence rate. Metrics for "Staff training" can be: hours of training per year per employee. Metrics for "noise" can be: percentage of workstations with noise levels exceeding 85 db. Metrics for "work accidents" can be: percentage of entities with periodically verified risk assessment, work-related incident rate.

Despite the fact that the number of social KPIs is greater than the environmental and economic indicators, social indicators need more intensive work to define the related

specific metrics, mainly due to the less experience of companies in this area. The selection identified includes all the parameters that every company should consider. During the analysis of the papers, a clear disparity in the use of environmental and economic performance indicators, rather than social indicators, was highlighted. Economic and environmental aspects were investigated in most of the articles analyzed. This did not happen for the social sphere, in fact, many companies, even today, have not implemented aspects related to the social sphere in their plan. For this reason, the emphasis was placed on social impact, identifying those parameters that every company should take into consideration in order to be defined as resilient and virtuous. After having enclosed the social indicators in macro-categories, 48 key indicators have been selected that will become part of the set of KPIs presented by the research.

There is a challenge that is emerging and involves all companies eager to become socially sustainable in industrial processes. These aspects are often misunderstood as the company focus is still focused on the economic and environmental aspects of I4.0.

## 5. Conclusions

The purpose of this paper was to define the state-of-the-art of how sustainability can be assessed and measured in different I4.0 industrial manufacturing contexts, considering its three areas (economic, environmental, and social). The paper presented an overview of sustainability KPIs based on the Triple Bottom Line concept, which refers to the three sustainability areas (social, economic, and environmental). Thanks to such KPIs (48 social indicators, 30 environmental indicators, and 39 economic indicators), companies could be able to measure corporate sustainability performance, providing data based on which strategic plans can be implemented. In this direction, the modern digital transition and I4.0 technologies can help companies to define and implement sustainability by correlating production and metrics. The article responds to this emerging gap by proposing a set of generic KPIs that can be used in any industrial context (regardless of company size and production). Particular attention was paid to social and organizational indicators as, to date, they are rarely implemented within manufacturing companies. To do this, it is necessary to integrate I4.0 technologies into various industrial contexts. In fact, they make it possible to obtain data in real time, exploit them and carry out actions to improve corporate sustainability. Eco-design makes it possible to predict environmental, social, and economic performance and allows strategic decisions to be made between possible industrial solutions. Though it has been highlighted that Industry 4.0 technologies are needed for data monitoring, the research highlighted the complete lack of such indicators. The authors have not found anything specific for this area (which takes into account the technologies of I4.0), so much so as to define this emerging gap as a starting point for future research. Although a detailed analysis of the state of the art of literature was performed, the selection of KPIs was limited starting from a specific year. Therefore, articles presenting other sustainability indicators may have been excluded from the analysis. Furthermore, the selection of the KPIs and the grouping into macro-categories were carried out personally by the authors, therefore a subjective evaluation was made. A final limitation of the research is that there are no case studies that demonstrate the effective validity of the set of KPIs in an industrial application and only for the social indicators have some metrics been defined in more detail to measure the KPIs.

Future research will concern a validation of the proposed set of KPIs by applying them in a specific business context, where a set of related metrics will be defined and implemented, and the three areas of sustainability will be correlated.

**Author Contributions:** Conceptualization, M.P.; methodology, M.P. and G.C.; validation, M.P.; data curation, G.C.; writing—original draft preparation, G.C.; writing—review and editing, M.P.; supervision, M.P. All authors have read and agreed to the published version of the manuscript.

**Funding:** This research was co-funded by SACMI SC.

**Institutional Review Board Statement:** Not applicable.

**Informed Consent Statement:** Not applicable.

**Data Availability Statement:** Not applicable.

**Conflicts of Interest:** The authors declare no conflict of interest.

## Appendix A

Tables A1–A3 list the articles presenting each macro-category of selected indicators (divided into social, environmental, and economic).

**Table A1.** List of articles presenting the selected Social KPIs.

| Social KPIs | No. Papers | Reference Papers |
|---|---|---|
| Health | 18 | [34] [53] [32] [66] [59] [65] [5] [22] [13] [62] [35] [12] [2] [23] [55] [14] [28] [22] |
| Human resources | 4 | [34] [48] [35] [23] |
| Human rights | 6 | [53] [32] [28] [64] [62] [23] |
| Ethics indicators | 2 | [16] [28] |
| Physical load index | 1 | [40] |
| Noise | 7 | [40] [59] [19] [5] [13] [43] [14] |
| Risk | 2 | [40] [13] |
| Wage | 5 | [40] [32] [19] [65] [28] |

**Table A1.** *Cont.*

| Social KPIs | No. Papers | Reference Papers |
|---|---|---|
| Workload | 2 | [40]<br>[48] |
| Injuries | 3 | [40]<br>[32]<br>[19] |
| Social investment | 1 | [40] |
| Fight against corruption | 4 | [40]<br>[32]<br>[62]<br>[23] |
| Workforce | 1 | [40] |
| Corrosive chemicals | 1 | [40] |
| Toxic chemicals | 1 | [40] |
| OSHA citations | 1 | [40] |
| Employee turnover | 5 | [40]<br>[28]<br>[2]<br>[43]<br>[28] |
| Employee satisfaction | 12 | [16]<br>[40]<br>[53]<br>[48]<br>[66]<br>[19]<br>[28]<br>[22]<br>[12]<br>[55]<br>[43]<br>[28] |
| Community quality of life | 1 | [40] |
| Community outreach activities | 1 | [40] |
| Charitable Contributions | 1 | [40] |
| Justice | 4 | [53]<br>[48]<br>[32]<br>[22] |
| Fair trading | 6 | [16]<br>[40]<br>[53]<br>[32]<br>[62]<br>[23] |
| Public service | 1 | [53] |
| Equal employment opportunities | 1 | [48]<br>[30] |
| Equal promotion opportunities | 1 | [48] |

**Table A1.** *Cont.*

| Social KPIs | No. Papers | Reference Papers |
|:---:|:---:|:---:|
| Payment ratio | 2 | [48] [43] |
| Complaints system | 1 | [48] |
| Staff training | 10 | [40] [65] [28] [18] [12] [2] [30] [63] [14] [28] |
| Child labor | 5 | [16] [32] [65] [62] [23] |
| Forced labor | 3 | [32] [65] [23] |
| Gender equity | 4 | [32] [19] [30] [14] |
| Social responsibility | 2 | [32] [23] |
| Product responsibility | 4 | [53] [22] [64] [21] |
| Community engagement | 2 | [32] [13] |
| Access to resources | 3 | [32] [62] [23] |
| Work accidents | 9 | [40] [48] [19] [28] [18] [30] [55] [63] [14] |
| Work illnesses | 4 | [40] [18] [12] [43] |
| Voluntary programs | 1 | [13] |
| Protection of private life | 2 | [62] [23] |

**Table A1.** *Cont.*

| Social KPIs | No. Papers | Reference Papers |
|---|---|---|
| Work conditions | 2 | [62]<br>[55] |
| Diversity | 2 | [40]<br>[12] |
| Community projects | 6 | [16]<br>[66]<br>[19]<br>[30]<br>[63]<br>[43] |
| Labor relationship | 3 | [28]<br>[13]<br>[12] |
| Philanthropy | 3 | [34]<br>[65]<br>[28] |
| Discrimination | 6 | [48]<br>[19]<br>[65]<br>[28]<br>[62]<br>[23] |
| Fatalities | 3 | [32]<br>[19]<br>[28] |

**Table A2.** List of articles presenting the selected Environmental KPIs.

| Environmental KPIs | No. Paper | Reference Paper |
|---|---|---|
| Global warming | 6 | [34]<br>[16]<br>[32]<br>[65]<br>[13]<br>[35] |
| Ozone depletion | 3 | [34]<br>[16]<br>[22] |
| Acidification | 4 | [34]<br>[16]<br>[65]<br>[43] |
| Eutrophication | 4 | [34]<br>[16]<br>[65]<br>[43] |
| Photochemical ozone | 4 | [34]<br>[16]<br>[65]<br>[43] |
| Resource use | 1 | [16] |

**Table A2.** *Cont.*

| Environmental KPIs | No. Paper | Reference Paper |
|:---:|:---:|:---:|
| Fuel use | 3 | [28] [61] [14] |
| Non-fossil resources | 1 | [34] |
| Fossil resources | 1 | [34] |
| Raw materials | 11 | [34] [40] [59] [19] [28] [22] [35] [2] [61] [43] [14] |
| Packaging materials | 1 | [59] |
| Consumables | 1 | [34] |
| Energy | 16 | [16] [40] [53] [32] [59] [19] [22] [13] [64] [18] [12] [2] [61] [63] [43] [14] |
| Transportation | 2 | [40] [48] |
| Biodiversity | 3 | [34] [62] [22] |
| Toxicity | 4 | [16] [32] [66] [65] |
| Waste | 13 | [16] [40] [53] [48] [66] [59] [13] [64] [30] [63] |

**Table A2.** *Cont.*

| Environmental KPIs | No. Paper | Reference Paper |
|---|---|---|
| | | [43] |
| | | [28] |
| | | [22] |
| Water | 19 | [40] |
| | | [53] |
| | | [59] |
| | | [19] |
| | | [65] |
| | | [5] |
| | | [28] |
| | | [22] |
| | | [64] |
| | | [62] |
| | | [18] |
| | | [35] |
| | | [12] |
| | | [21] |
| | | [2] |
| | | [23] |
| | | [55] |
| | | [63] |
| | | [43] |
| Durability | 2 | [16] |
| | | [43] |
| Greenhouse gas | 8 | [40] |
| | | [53] |
| | | [59] |
| | | [28] |
| | | [22] |
| | | [64] |
| | | [12] |
| | | [43] |
| Renewable energy | 7 | [59] |
| | | [62] |
| | | [12] |
| | | [21] |
| | | [23] |
| | | [63] |
| | | [43] |
| Air emission | 10 | [53] |
| | | [66] |
| | | [19] |
| | | [65] |
| | | [5] |
| | | [22] |
| | | [13] |
| | | [55] |
| | | [61] |
| | | [14] |
| Hazard materials | 6 | [53] |
| | | [66] |
| | | [59] |
| | | [22] |
| | | [43] |
| | | [28] |

**Table A2.** *Cont.*

| Environmental KPIs | No. Paper | Reference Paper |
|---|---|---|
| Recycling | 8 | [59] [13] [62] [12] [21] [30] [23] [28] |
| Pollution | 6 | [53] [22] [62] [21] [23] [43] |
| Radioactive emissions | 1 | [59] |
| EOL product | 1 | [59] |
| Climate change | 1 | [65] |
| Land use | | [53] [22] [62] [35] [14] [28] |
| Noise | 3 | [59] [43] [14] |

**Table A3.** List of articles presenting the selected Economic KPIs.

| Economic KPIs | No. Paper | Reference Paper |
|---|---|---|
| Manufacturing costs | 5 | [34] [40] [53] [48] [62] |
| Commercial costs | 1 | [34] |
| Research and development costs | 8 | [34] [59] [19] [28] [62] [18] [2] [28] |
| General and administrative costs | 1 | [34] |
| Financial costs | 4 | [34] [16] [32] [23] |

**Table A3.** *Cont.*

| Economic KPIs | No. Paper | Reference Paper |
|---|---|---|
| Environmental costs | 4 | [34] [65] [28] [13] |
| Social costs | 2 | [34] [21] |
| Ethical investments | 4 | [16] [21] [63] [43] |
| Health and safety costs | 3 | [19] [21] [43] |
| Net profit | 3 | [40] [65] [18] |
| Facility costs | 4 | [40] [32] [19] [65] |
| Labor costs | 7 | [40] [48] [59] [19] [65] [61] [14] |
| Material costs | 8 | [40] [53] [32] [66] [59] [65] [62] [14] |
| Utility costs | 1 | [40] |
| Eco friendly investment | 4 | [19] [65] [13] [2] |
| Warranty costs | 1 | [59] |
| Return on assets | 2 | [40] [28] |
| Delivery performance | 3 | [40] [48] [14] |
| Logistics costs | 3 | [53] [59] [30] |
| End of life costs | 2 | [53] [13] |

**Table A3.** *Cont.*

| Economic KPIs | No. Paper | Reference Paper |
|---|---|---|
| Energy costs | 6 | [53]<br>[66]<br>[59]<br>[65]<br>[62]<br>[35] |
| Packaging costs | 4 | [53]<br>[59]<br>[19]<br>[43] |
| Scrap costs | 3 | [53]<br>[66]<br>[19] |
| Water costs | 1 | [53] |
| Maintenance costs | 5 | [53]<br>[48]<br>[19]<br>[65]<br>[62] |
| Cost of PPE | 2 | [53]<br>[19] |
| Training costs | 2 | [53]<br>[59] |
| Costs of waste disposal treatment | 2 | [53]<br>[66] |
| Lead time | 4 | [53]<br>[48]<br>[19]<br>[61] |
| Transportation costs | 1 | [59] |
| Turnover | 3 | [19]<br>[28]<br>[18] |
| Staff costs | 1 | [28] |
| Sales | 8 | [32]<br>[66]<br>[59]<br>[19]<br>[28]<br>[13]<br>[30]<br>[63] |
| Supply chain costs | 4 | [48]<br>[64]<br>[62]<br>[63] |
| Remanufacturing | 1 | [62] |
| Recycling processes | 2 | [62]<br>[12] |
| Employee satisfaction | 3 | [13]<br>[55]<br>[43] |

**Table A3.** *Cont.*

| Economic KPIs | No. Paper | Reference Paper |
|---|---|---|
| Quality | 4 | [59] [19] [5] [55] |
| Taxes | 4 | [32] [19] [30] [63] |

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
