# Peer review of "Sustainability and Industry 4.0: Definition of a Set of Key Performance Indicators for Manufacturing Companies"

_sustainability, doi:10.3390/su141711004_

Round 1
Reviewer 1 Report
Dear authors,
Thanks a lot for giving me the possibility to review the paper named "Sustainability and Industry 4.0: Definition of a Set of Key Performance Indicators for Manufacturing Companies". I think that the topic is exciting for the research community.
About the paper :
a) Introduction and literature: I think the context is not well defined or the title is not suitable; I find the paper more a part of literature than an analysis (it needs to be improved- I don't suggest any title; it's not my cup of tea). The title of the paper is not suitable for the paper research.
b) methodology: I do not find the definition of key performance indicators. I find a good analysis of possible variables to be connected or combined, but I do not find key performance indicators. How will you carry them out? We need something that measures....
c) How will you test (after defining) them? Industry 4.0 is well-known, everybody has written, but little is done or is well designed.
d) about conclusions: it's too generic; we need to understand more about the weakness of your research paper and a good (possible) route for research.
Reviewer 2 Report
Name of the Paper: Sustainability and Industry 4.0: Definition of a Set of Key Performance Indicators for Manufacturing Companies
The paper attempts to review articles to define a set of KPIs for Manufacturing Industry 4.0. The Triple Bottom Line concept was applied to identify sustainability KPI. The company may use the identified sustainability KPI for gauging its sustainability level.
General Observation:
1) Please refer to "three areas" they are mentioned at the end of the manuscript. Please discuss these three areas in the introduction section with reference to sustainability.
2) Please refer to Figure 1. Methodology for the review paper selection, merely mentioning -Topic, -Type of document doesn't provide clarity. Please amend the figure to provide more details. There is a pink dash line appearing along the 8704 Contribution which may be removed.
3) Please refer to "Table 1. Selected paper published in different journals and related classes of KPIs", the paper title is poorly mentioned, and there is a mix up of small and capital letters. The first title shows 'Ap-proach'
4) Please refer to "Table 1. Under the caption “Paper”, the papers are not arranged, there are some papers indicating 2022, The timeline selected is not clear.
5) Please refer to "Table 1. Under the caption “Paper”, the paper title with analysis is also found.
6) The papers are not arranged, there are some papers indicating 2022, The timeline selected is not clear.
7) Please refer to “for each filter, a certain number of articles have been selected” this is misleading.
8) Please refer to "Table 1. Almost all papers are classified under "three areas". Moreover, it is merely a list of identified papers. Please draw some significant findings from such contributions to provide value addition to Table 1.
9) Please refer to “Table 1 highlights the lack of a single set of generic KPIs to measure sustainability in a manufacturing company.” It is unclear how this inference is drawn.
10) Please refer to "Definition of a Set of Key Performance Indicators for Manufacturing Companies", the Table 2 KPI of the selected articles shows a mixed group not restricting finding to manufacturing companies only which is contradicting the aim of the study For instance ‘Oil and Gas Industry.’
11) Please refer to Table 2. There is information flooding on KPI of the selected articles, It is hardly readable for meaningful takeaway.
12) Please refer to Table 2. The first column of the table shows the type of industry and reference to it (Generic Company (Peruzzini and Pellicciari,2018)) but later on, this is not continued, only the reference article is mentioned (Yildiz Çankaya and Sezen, 2019), (Shuaib et al., 2014), (Cagno et al., 2019) so on till the end.
13) Figure 2. Social Indicators, Figure 3. Environmental Indicators and Figure 4. Economic Indicators are hardly readable and look over propionate. Authors may amend this or provide meaningful information in tabular form.
14) Industry 4.0 is on internet-based technology (like deep learning, IoT, Cloud computing, robots, etc.), and there are no KPIs related to it. How the authors justify this to its title of “Sustainability and Industry 4.0:”
Reviewer 3 Report
The paper presents an interesting and up-to-date topic of defining a set of Key Performing Indicators for manufacturing companies. The article's topic corresponds well with the scope of the Sustainability journal of MDPI. The paper presents the theoretical background and a thorough literature review on the topic.
The paper present a good academic level however there are a few comments to improve and to organize the paper better:
1. Please provide the research questions or the hypothesis for the paper at the beginning.
2. The reviewing paper selection consist of 63 papers, the total amount of references is 66, in the theoretical introduction no other papers were referred to. Consider extending the reference list.
3. Please check the referencing section. Not all papers are accurately cited
4. Fig. 1. Please improve the workflow to better suit the journal needs
5. Please consider if tables 3-5 are needed in the paper (maybe they can be moved to the appendix, and just the figures 2-4 are enough in the text?
6. Please detach results and discussion as two separate chapters; right now, the 3.2. the chapter is too mixed and too blurry. Please focus on this part since, as the most essential part of the paper, it does not present the results clearly and understandably. Some parts of this chapter should be also moved to conclusions as the general information summarizing the paper.
Round 2
Reviewer 1 Report
All I suggested have bene done.
Reviewer 2 Report
Some of the comments are yet to be addressed carefully.
1) Please refer to "Table 1. Selected paper published in different journals and related classes of KPIs", The authors need to separate the title and journal titles as per the table caption.
For example paper number 1: M. Abubakr, A. T. Abbas, I. Tomaz, M. S. Soliman, M.Luqman, and H. Hegab, ‘Sustainable and Smart Manufacturing:
An Integrated Approach, Sustainability, vol. 12, no.6, p. 2280, Mar. 2020, doi: 10.3390/su12062280.
It contains the name of the journal/DOI and the last column also represents the name of the journal. So it may be updated accordingly.
At present, Table 1 does not add any values except the classification into three categories. Authors may review the papers to identify the KPIs to classify into a meaningful classification for instance quantitative or qualitative, Focus of KPI, Number of KPIs, countrywide KPIs, etc.
2) Table 2. KPI of the selected articles (29) doesn't correlate completely with the 63 articles, further, the basis of selection is unclear.
3) The inference drawn from Table 1 and Table 2 may be highlighted
4) Few KPIs occur in more than one category of Social, environmental, and Economic.
5) Authors may also discuss the role of triple bottom line companies in Industry 4.0 in terms of Sustainability.
Reviewer 3 Report
Thank you for addressing all my comments. In this form, the paper presents a higher scientific value. For this reason, I recommend its publication.